# Oral *Lactobacillus* Species in Systemic Sclerosis

**DOI:** 10.3390/microorganisms9061298

**Published:** 2021-06-15

**Authors:** Daniela Melchiorre, Maria Teresa Ceccherini, Eloisa Romano, Laura Cometi, Khadija El-Aoufy, Silvia Bellando-Randone, Angela Roccotelli, Cosimo Bruni, Alberto Moggi-Pignone, Davide Carboni, Serena Guiducci, Gemma Lepri, Lorenzo Tofani, Giacomo Pietramellara, Marco Matucci-Cerinic

**Affiliations:** 1Department of Experimental and Clinical Medicine, Department of Geriatric Medicine, Division of Rheumatology, University of Firenze, 50124 Firenze, Italy; eloisaromano@libero.it (E.R.); lauracometi@gmail.com (L.C.); khadija.ela92@gmail.com (K.E.-A.); silvia.bellandorandone@unifi.it (S.B.-R.); cosimobruni85@gmail.com (C.B.); alberto.moggipignone@unifi.it (A.M.-P.); carbonidavide10@gmail.com (D.C.); serena.guiducci@unifi.it (S.G.); gemma.lepri@gmail.com (G.L.); lorenzo120787@gmail.com (L.T.); marco.matuccicerinic@unifi.it (M.M.-C.); 2Department of Agriculture, Food, Environment and Forestry (DAGRI)-University ofFirenze, 50144 Firenze, Italy; mariateresa.ceccherini@unifi.it (M.T.C.); angela.roccotelli@unifi.it (A.R.); giacomo.pietramellara@unifi.it (G.P.)

**Keywords:** oral microbiome, *Lactobacillus spprpoB* gene, SSc, qPCR, quality of life

## Abstract

In systemic sclerosis (SSc), the gastrointestinal tract (GIT) plays a central role in the patient’s quality of life. The microbiome populates the GIT, where a relationship between the *Lactobacillus* and gastrointestinal motility has been suggested. In this study, the analysis of oral *Lactobacillus* species in SSc patients and healthy subjects using culture-independent molecular techniques, together with a review of the literature on microbiota and lactobacilli in SSc, has been carried out. Twenty-nine SSc female patients (mean age 62) and twenty-three female healthy subjects (HS, mean age 57.6) were enrolled and underwent tongue and gum swab sampling. Quantitative PCR was conducted in triplicate using *Lactobacillus* specific primers *rpoB*1, *rpoB*1o and *rpoB*2 for the RNA-polymerase β subunit gene. Our data show significantly (*p* = 0.0211) lower *Lactobacillus*spp*rpoB* sequences on the tongue of patients with SSc compared to HS. The mean value of the amount of *Lactobacillus ssprpoB* gene on the gumsofSSc patients was minor compared to HS. A significant difference between tongue and gums (*p* = 0.0421) was found in HS but not in SSc patients. In conclusion, our results show a lower presence of *Lactobacillus* in the oral cavity of SSc patients. This strengthens the hypothesis that *Lactobacillus* may have both a protective and therapeutic role in SSc patients.

## 1. Introduction

The pathogenesis of systemic autoimmune diseases, including systemic sclerosis (SSc), is complex and characterized by auto-reactive immune responses that cause immune-mediated organ damage [1,2,3,4]. Indeed, despite the fact that pathogenetic aspects have not yet been completely elucidated, environmental factors (lifestyle, diet, drugs and infections) acting on people with permissive genetic assets have been proposed [2].

Among the different environmental factors, one that stands out is the microbiota, which is the whole composition of microorganisms, mainly bacteria, but even fungi and viruses, that densely populate the human body [5,6]. They represent about 90% of the cells associated with any human being [6], and their genome and ecosystem are called the microbiome. It is assumed that a minimum of 1000 species are involved [2], with extreme variability between individuals and also within each individual, depending on the body site sampled and the time of sampling [7]. The microbiome occupies different habitats, such as skin and mucosal surfaces. The great majority is found in the gastrointestinal tract (GIT) and the predominant phyla are Bacteroidetes and Firmicutes [6,8]. The GIT microbiota plays a pivotal role in the immune system’s development, homeostasis, and function. Emerging evidence suggests that GIT dysbiosis (changes in the composition of the microbiota, such as the loss of beneficial microorganisms, excessive growth of potentially harmful organisms, loss of overall microbial diversity) leads to a loss of tolerance and the subsequent development of a rheumatic disease, including rheumatoid arthritis (RA), spondyloarthritis, Sjögren’ssyndrome and SSc [2,5,8,9].

In SSc, GIT involvement is one of the key features with a great impact on the patient’s quality of life [9]. Symptoms of lower GIT involvement, such as constipation, abdominal pain, diarrhea, fecal incontinence, malabsorption, and weight loss, are among the more disruptive physical problems and have a great impact on the SSc patient. The alterations in peristalsis may favor dysbiosis and systemic bacterial translocation, both of which have the potential to alter systemic immune responses, and the vasculopathy may alter mucosal barrier function, integrity, and gut homeostasis [1]. Furthermore, small intestinal bacterial overgrowth is a well described complication associated with GI dysmotility, discomfort and malnutrition [10]. However, the etiology of SSc-related lower GI dysfunction is largely unknown, and no effective treatment options are currently available.

Observational studies have demonstrated that *Lactobacillus*, frequently associated with *Bifidobacterium*, was recognized to be increased at gut level in SSc patients, two commensal genera typically less present in chronic inflammatory status [8,11,12,13,14]. In particular, *Lactobacillus* was found to be greater in patients with limited than in diffuse cutaneous sclerosis [12]. An increase in colonic *Lactobacillus* has also been observed in different cohorts, and a link between members of the *Lactobacillus* genus and gastrointestinal motility has been suggested.

The *Lactobacillus* genus is a relevant representative of the Firmicutes phylum. It is a very complex genus with several species, and it was recently reorganized from a taxonomic point of view by adding further species [15]. It is a normal constituent of the microbiota, even though only a minority of species are autochthonous to the human GIT, the large majority being considered allochthonous [16], and it has generally been associated with beneficial activity [14]; in fact, there are a lot ofstudies on its use as a probiotic [17] and even in health it seems to be associated with an anti-inflammatory activity [18], whereas only rarely it has been described to behave as an opportunistic microorganism [19]. Despite the fact that *Lactobacillus* has been successfully used as a probiotic/anti-inflammatory agent in experimental models of systemic immune-mediated inflammatory diseases since the end of the last century [20,21,22,23,24,25,26] and its presence has been associated with anti-inflammatory activity [27], in the saliva of primary Sjögren’ssyndrome [28] and in the gut of RA [29,30] and SSc, it has been repeatedly found at an increased level [11,12,13,14] and associated with GIT pathologic involvement [12], thus casting some doubts on the actual association with anti-inflammatory activity (Table 1). Conversely, nothing is known, to the best of our knowledge, on the role and the levels of oral *Lactobacillus* in SSc patients. It is well known that the oral microbiome deeply influences the gut microbiome, so that they are strictly interwoven. Moreover, the importance of the oral microbiome as a cause of both oral and systemic diseases was recently evaluated. This is due to the transformation of commensal bacteria into opportunistic bacteria inside the oral cavity, which are able to determine the onset of pathologic conditions either limited to the oral cavity, including caries, gingivitis, periodontitis, and tonsillitis, or systemic [31,32]. Clinical studies have associated periodontopathic bacteria with some systemic disorders, such as RA, which seem to have similar physio-pathological mechanisms. In fact, periodontal bacteria are reported to induce the production of neutrophil, monocyte and T and B-mediatedimmune responses and the release of proteinases, cytokines and prostaglandins, causing bone osteoclastactivity and bone erosion, similar to the pathophysiology of RA [33].

The aim of this study was to apply the molecular analysis of quantitative polymerase chain reaction (PCR) on oral *Lactobacillus* spp. in SSc patients and sex- and age-matched healthy controls.

## 2. Materials and Methods

Consecutive SSc outpatients of the Rheumatology Unit of AOU Careggi, Firenze, Italy, were enrolled in our study. Patients were classified according to ACR 2013 criteria and for each patient clinical and anamnestic data were collected (type of disease and type of organ involvement, antibody profile, therapies performed, PRO questionnaire (HAQ, VAS, UCLA-GIT)). Inclusion criteria were age >18 years old, diagnosis of SSc, absence of active periodontal disease or dental caries, absence of other systemic diseases, and a regular dietary regimen. The exclusion criteria were the recent use (1 month) of antibiotic/antifungal therapy, the use of probiotics, the presence of periodontal disease or dental caries and a particular dietary regimen. Healthy subjects (HS), who did not present active periodontal disease or dental caries and did not follow a particular dietary regimen, were also analyzed. 

Patients and HS signed an informed consent form to participate in the study, all procedures were carried out in accordance with the Helsinki Declaration of 1964 and the study was approved by the local ethical committee (15013/CAM_BIO).

The patients and HS included in the study were subjected to a sampling of tongue and gum swabs (sterile swabs with cap in polyethylene, inserted in tube in polypropylene with plastic shaft in dry transport media. Biosigma spa, Italy). The nurse (K.E.-A.) performed a gum swab in patients and HS by repeated rubbings on all surfaces of gingival tissue (internal and external surfaces of the gums, in the upper and lower part of the mouth). The tongue swab was performed by repeated rubbings of the swab over the entire lingual surface (on the lateral, upper and lower parts of the tongue). All samples were delivered to the laboratory and stored at −80 °C before DNA extraction.

### 2.1. DNA Extraction and Quantification of Lactobacillus

Total DNA was extracted from the swabs sampled by the nurse from SSc patients and HS by QIAampDNAStool Mini Kit instructions (Qiagen, S.r.l. Italy) and quantified with a Qubit^®^ 2.0 fluorometer (Invitrogen, Thermofisher Scientific, Milano, Italy); the qualityof DNA was checked on 1.5% agarose gel. Quantitative PCR (qPCR) was conducted using the specific primers *rpoB*1, *rpoB*1o and *rpoB*2 for the RNA-polymerase β subunit encoding gene, a valuable alternative as present in a single copy per genome, whichgenerate amplicons of 250-bp, on 40 ng DNA for all the samples; negative control contained only ddH2O. Reactions were performed in a CFX Connect 96 apparatus (BioRad, Hercules, CA, USA) and the results were analyzed by the manufacturer’s software. Amplification was carried out in a 25 μL final volume containing: 7.5 μM of each primer, 1X iTaq™ Universal SYBR^®^GreenSupermix, sterile ddH2O to reach the appropriate volume. Amplification was performed in 96-well microtiter plates (BioRad). The program cycle was: 95 °C 3 min followed by 35 cycles of 95 °C 1 min, 45 °C 1 min and 72 °C 1 min. After that, a melting curve program was run for which measurements were made at 0.5 °C temperature increments every 10 s within a range of 60–100 °C. A *rpoB* amplified and purified fragment (froma mix of LABs of a commercial probiotic formulation—Lactoflorene^®^ Plus, Montefarmaco OTC, Milano, Italy) was used as standard [34]. The standard curve was developed by plotting the logarithm of known concentrations (tenfold dilution series in triplicate from 1 × 10^−1^ to 1 × 10^−6^ in 25 μL reaction) of the *rpoB* fragment against the threshold cycle (Ct) values. The qPCR standard curve had an R2 of 0.99–0.97 and an efficiency >85%. Three replicates were carried out for each sample. *rpoB*sequences were expressed in ng DNA ng−1 of template DNA.

### 2.2. Statistical Analysis

To evaluate the difference in *Lactobacillus* qPCR in gum and tongue between groups, a GEE (generalize estimating equation) linear regression model was used. Student’s *t* test for continuous variables was also used.

## 3. Results

### 3.1. Patients and Healthy Subjects

Twenty-nine consecutive SSc female outpatients (mean age ± standard deviation (SD) 62 ± 12.43; range 43–80 years) of the Rheumatology Unit of AOU Careggi, Firenze, Italy, were enrolled. Out of 29 patients, 13 showed limited cutaneous systemic sclerosis (lcSSc), 15 diffuse cutaneous systemic sclerosis (dcSSc), 1 overlap syndrome. There were seven current and nine former smokers. Twenty-three female and age-matched healthy subjects (HS, mean age ± SD 57.6 ± 8.46; range 44–72 years), seven of whom were current and fivewere former smokers, were also analyzed. There was no significant difference between patients and HS for smoking habits (Table 2).

LcSSc group: all patients had a mean disease duration of 12 years and showed anti-nuclear antibody (ANA) and anti-centromere antibody (ACA) positivity (one patient was also anti-RNA polymerase-3 positive). Six patients were on chemical Disease Modifying Anti-Rheumatic Drugs (cDMARD) therapy (two mycophenolate mofetil, four hydroxychloroquine), three patients were on i.v.prostanoid (iloprost), and one patient had previously been treated with cyclophosphamide. Smoking was reported in seven patients (four current smokers and three former smokers).

DcSSc group: all patients had a mean disease duration of 14 years and showed ANA and anti-Scl70 positivity (one patient was also anti-RNA polymerase-3 positive). Ten patients were receiving cDMARD (two corticosteroids, two azathioprine, one methotrexate, five hydroxychloroquine) and three of them were taking IVIG, 10 patients were on i.v.prostanoid (seveniloprost, three alprostadil), and two patients had been previously treated with cyclophosphamide. Three were current smokers and six were former smokers.

One patient presented with overlap syndrome characterized by ANA, anti-Scl70, anti-SSA, and anti-SSB positivity with a long-lasting disease (22 years) and was on hydroxychloroquine and golimumab (Table 2).

### 3.2. Lactobacillus rpoB Quantification in Tongue Samples

The mean value of the amount of *Lactobacillus* spp. *rpoB* gene in tongue samples of SSc patients was 4.14 × 10^−^^6^ (±88 × 10^−6^) ng ng^−1^ DNA.The mean value of the amount of *Lactobacillus* spp. *rpoB* gene of HS was 7.09 × 10^−6^ (±7.21 × 10^−6^) ng ng^−1^ DNA. A significant difference was detected between the two groups (*p* = 0.0211).

### 3.3. Lactobacillus rpoB Quantification in Gum Samples

The mean value of the amount of *Lactobacillus* spp. *rpoB* gene in gum samples was 3.55 × 10^−6^ (±3.72 × 10^−6^) ng ng^−1^DNA in SSc patients. The mean value of the amount of *Lactobacillus* spp. *rpoB*in HS was 3.72 × 10^−6^ (±3.93 × 10^−6^) ng ng^−1^ DNA. No significant differences were detected between the two groups.

The *rpoB* sequence number per ng of template DNA quantified by qPCR was significantly different between tongue and gum (*p* = 0.0421) in HS, whereas no significant differences were detected in SSc patients (Figure 1).

## 4. Discussion

The possibility that the microbiome may influence the onset and the severity of autoimmune diseases has become a field of active studies in the last few years. Teodorescu et al. have reported that chronic periodontitis can be more frequent in adults. This periodontal destruction could be directly correlated with biofilm and calculus deposits, the progression rate is slow to moderate and the patient can also present local and general risk factors as systemic diseases [35]. The molecular methods more precisely reflect the true microbial local situation, considering that their sensitivity compared to the traditional cultural methods is much higher [34,36]. However, in spite of all these technical developments and the net increase in the relative studies, only a few certainties have been reached up to now. The role of *Lactobacillus* species is controversial (Table 1). This is also true for other rheumatic diseases in which an increase in the quantity of *Lactobacillus* has been found [17,18,37,38,39,40,41], but more recently the opposite has been observed [12].

Conversely, *Lactobacillus* has been found consistently increased in the gut microbiota of patients with SSc in studies from the US, Italy and Norway [11,12,13,14,15,16,17,18,19,20,21,22,23,24,25,26,27,28,29,30,31,32,33,34,35,36,37,38,39,40,41,42]. *Lactobacillus* and *Bifidobacterium*, two commensal genera, are typically less present in the chronic inflammatory status, and *Lactobacilla* were found to be greater in patients with limited cutaneous sclerosis than in diffuse cutaneous sclerosis [8]. These aspects cast some doubts on the protective role of *Lactobacillus* ssp. in SSc, as it appears to be so in RA. However, nothing is known, to the best of our knowledge, on the presence of oral *Lactobacillus* in patients with SSc, in spite of the relevance of this information, considering its possible local anti-inflammatory activity, a body district which is profoundly involved in SSc patients, and the possibility of modulating the *Lactobacillus* presence through dietary and/or drug interventions. Smoking does not seem to influence the oral *Lactobacillus*, considering that *Lactobacillus* is only enriched in currentsmokers [43], who were not different in SSc patients compared with HS. Our data show a significant difference in *Lactobacillus* spp. *rpoB* gene levels between HS and SSc patients. *Lactobacillus* spp. *rpoB* gene levels are much lower on the tongues of SSc patients, suggesting the involvement of oral *Lactobacillus* in the SSc local modulation. Although microbial abundance in gut microbiota is generally considered dependent on the levels observed in the oral microbiota, this does not seem to be the case for *Lactobacillus* in SSc patients, perhaps for a number of reasons largely unknown, but which may allow us to hypothesize that the oral cavity of SSc patients may be quite an inhospitable environment for *Lactobacillus*, which may find a more welcoming survival niche at the colonic mucosa. *Lactobacillus* species offer exciting research opportunities, both in terms of biomedical applications and in acquiring fundamental knowledge about the functionality of gut microbes. It is also important to underline that lactic bacteria, due to their pro-bioactive cellular substances, are considered probiotic agents with an important role in the human microbiome and are not involved in oral diseases [34,44,45,46,47,48,49,50,51,52,53,54,55,56]. Their use could improve the quality of life of rheumatic patients, and in particular the possible therapeutic role of *Lactobacillus* in patients suffering from SSc or other rheumatic diseases is suggestive. The bacteria residing in the mammalian gut and their hosts are likely to have coevolved over a long-conjoined history and, by doing so, have developed an intimate and complex symbiotic relationship. The mechanisms underlying these interactions are likely to be specific for a particular microbe and its host and are probably influenced by other partners of the gut microbiota. Therefore, investigations of the host/microbe interplaying gut ecosystems should be conducted within an ecological context. It is reasonable to consider that native strains are better probiotic strains for some applications [57,58]. It has been clearly shown that gut microbes benefit their host in many aspects. In fact, gut bacteria can enhance host immune functions and the mucosal barrier, and they provide protection against incoming microbes [11,59,60,61,62,63,64,65,66].

Our investigation aimed to explore the potential protective role of oral *Lactobacillus* in SSc patients, considering that reduced microbial richness found in SSc gut was associated with a worse prognosis, such as a diffuse skin subset, poor nutritional status or a longer duration of the disease [14].

Our data show that the number of *Lactobacillus* on the tongue of patients is about half that of HS. Moreover, the statistical analysis performed showed a significant difference between the *Lactobacillus* present on the gums and those present on the tongue of HS in contrast to what was obtained in SSc patients, in whom no significant difference was found.

It is noteworthy that this is the first time, to the best of our knowledge, that a statistically significant reduction in *Lactobacillus* in the oral cavity of SSc patients has been demonstrated. This allows us to hypothesize the possible protective role of *Lactobacillus*, in line with its well-known association with anti-inflammatory activity [21], and consequently possible intervention strategies, based on *Lactobacillus*-containing probiotic administration, in an autoimmune disease in which the available therapeutic tools are currently very poor.

## Figures and Tables

**Figure 1 microorganisms-09-01298-f001:**
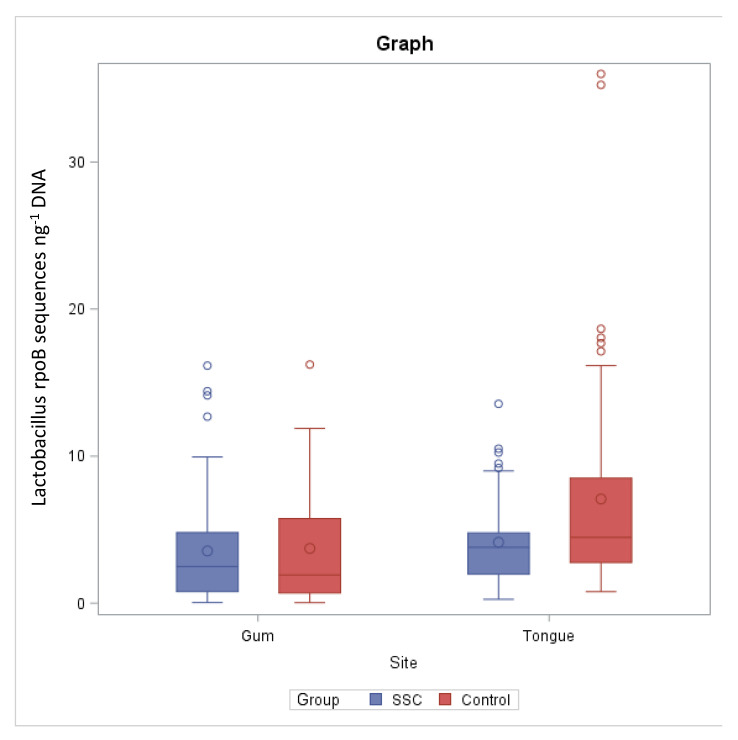
*Lactobacillus* spp. *rpoB* sequences (expressed as ng^−1^ of target DNA) quantified by qPCR in gums and tongues of SSc patients and healthy subjects (control).

**Table 1 microorganisms-09-01298-t001:** Controversial role of *Lactobacillus* spp. in systemic immune-mediated inflammatory diseases.

Species	Quantity/Function	Reference
*Lactobacillus casei*	Prevention of diabetes mellitus in NOD mice	[20]
*Lactobacillus delbrueckii subsp. bulgaricus*	Prevention of collagen-induced arthritis in mice	[21]
*Lactobacillus GG*	Oral intake improves arthritis in Lewis rats	[22]
*Lactobacillus casei*	Potentiated induction of oral tolerance in EAA	[23]
*Lactobacillus paracasei/Lactobacillus plantarum*	Improvement of EAE, by Treg 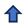 and Th1/Th17 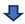	[24,25]
*Lactobacillus* spp.	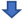 in the gut of lupus-prone mice. Restoring normal number reverse SLE	[26]
*Lactobacillus* spp.	Increase in saliva of primary Sjögren’ssyndrome patients	[28]
*Lactobacillus* spp.*/Lactobacillus salivarius*	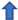 in RA patients’ gut, possible link with disease onset/progression	[29,30]

NOD = non-obese diabetic; EAA = experimental autoimmune arthritis; EAE = experimental autoimmune encephalomyelitis; Treg = T regulatory cells; Th1/Th17 = T helper 1/T helper 17 cells; SLE = systemic lupus erythematosus; RA = rheumatoid arthritis.

**Table 2 microorganisms-09-01298-t002:** Demographic, lifestyle and clinical characteristics of study patients.

Characteristic	SScPatients (*n* = 29)	Healthy Subjects (*n* = 23)
**Age (years)** Mean ± SD Range	62 ± 12.43 43–80	57.6 ± 8.47NS 44–72
**Female gender**	29	23
**Smoking** current former never	7 9 13	7NS 5NS 11NS
***LcSSc***	13	NA
Disease duration (years)	12	NA
ANA positivity	13 (100%)	NA
ACA positivity	13 (100%)	NA
***DcSSc***	15	NA
Disease duration (years)	14	NA
ANApositivity	15 (100%)	NA
Anti-scl70 positivity	15 (100%)	NA
***Overlapsyndrome***	1	NA
Disease duration (years)	22	NA
ANA, anti-scl70, SSA, SSB	Positive for all	NA
cDMARDs	16 (58.6%)	NA
I.v.prostanoid	13 (44.8%)	NA

LcSSc = limited cutaneous systemic sclerosis; DcSSc = diffuse cutaneous systemic sclerosis; ANA = anti-nuclear antibodies; ACA = anti-centromere antibodies; cDMARDs = chemical Disease Modifying Anti-Rheumatic Drugs; NS = not significant; NA = not applicable.

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
