# Peer review of "Oral Lactobacillus Species in Systemic Sclerosis"

_microorganisms, 2021, doi:10.3390/microorganisms9061298_

Round 1

Reviewer 1 Report

This is a very interesting, well thought study that has much scientific merit, however a few issues must be adressed before publication.

Abstract

 Line 21: together with are view – a review

 "Twenty-nine SSc female patients (mean age 62.) and 23 female healthy subject" – choose number or word for consistency

Line 23 is not necessary “Total DNA was extracted from 23 swabs by QIAamp DNA Stool Mini Kit (Qiagen).”

From line 26 rephrase too many details, not enough significance.

No conclusions in the abstract, please include

Introduction

The role of Lactobacillus (in disease or health) in the oral microbiome is not clearly stated. I suggest elaborating in it's role in the developement of dental caries and periodontal disease. These citations are a good fit for that

  1. Martu MA, Solomon SM, Sufaru1 IG, Jelihovschi I, Martu S, Rezus E, Surdu AE, Onea RM, Grecu GP, Foia L. . Study on the prevalence of periodontopathogenic bacteria in serum and subgingival bacterial plaque in patients with rheumatoid arthritis. Rev. Chim. (Bucharest). 2017; 68(8): 1946-1949
  2. Solomon SM, Bataiosu M, Popescu DM, Rauten AM, Gheorghe DN, Petrescu RA, Maftei GA, Maglaviceanu CF. Biochemical Assesment of Salivary Parameters in Young Patients with Dental Lesions. Revista de Chimie. 2019 Nov 1;70(11):4095-7

Materials and Methods

  • did the patients have other systemic pathologies?
  • what was the periodontal and dental status of patients?
  • what was the diet of patients, since it can have a significant role in SSc

Discussions: 

In the introduction the authors mentioned a great implication of Fusobacterium, Prevotella and  γ-Proteobacteria in the pathogeny of SSc, however these bacteria were not analysed, instead another one was chosen for this study, why is that?

In the discussions section the authors should emphasize the role of oral microbial shifts in health and disease, i suggest adding these articles:

  1. Assessment of Bacterial Associations Involved in Periodontal Disease Using Crevicular Fluid. Teodorescu AC, Teslaru S, Solomon SM, Zetu L, Luchian I, Sioustis IA, Martu MA, Vasiliu B, Martu S. Chim.(Bucharest), 2019 Jun 1;70(6):2145-9.
  2. Martu I., Goriuc A, Martu MA, Vata I, Baciu R, Mocanu R, Surdu AE, Popa C, Luchian I. Identification of bacteria involved in periodontal disease using molecular biology techniques. Rev. Chim.(Bucharest), 2017,68(10): 2407-2412.

Reviewer 2 Report

Thank you for submitting your manuscript to Microorganisms. 

General comments:

  1. The rationale / research question was not clear. As such, it was difficult to follow the flow..
  2. The introduction is too long and may in a large part not be pertinent to the paper objectives.

Specific comments:

  1. The methodology did not describe the swab source (vendor) , nor the method of swabbing. Swabbing was key to the study.
  2. Exclusion criteria did not list the exclusion of males.
  3. Given 1 figure in the results, paper should be re-written as a note
